# Functional Analysis of the Soybean *GmCDPK3* Gene Responding to Drought and Salt Stresses

**DOI:** 10.3390/ijms20235909

**Published:** 2019-11-25

**Authors:** Dan Wang, Yuan-Xia Liu, Qian Yu, Shu-Ping Zhao, Juan-Ying Zhao, Jing-Na Ru, Xin-You Cao, Zheng-Wu Fang, Jun Chen, Yong-Bin Zhou, Ming Chen, You-Zhi Ma, Zhao-Shi Xu, Jin-Hao Lan

**Affiliations:** 1College of Agronomy, Qingdao Agricultural University, Qingdao 266109, China; wangdanyx@stu.qau.edu.cn (D.W.); yuanxialiu@163.com (Y.-X.L.); qianstyle123@sina.com (Q.Y.); 2Key Laboratory of Biology and Genetic Improvement of Triticeae Crops, Ministry of Agriculture, Institute of Crop Sciences, Chinese Academy of Agricultural Sciences (CAAS)/National Key Facility for Crop Gene Resources and Genetic Improvement, Beijing 100081, China; zhaoshuping001@163.com (S.-P.Z.); zjy0502@yeah.net (J.-Y.Z.); rujingna1993@163.com (J.-N.R.); chenjun01@caas.cn (J.C.); zhouyongbin@caas.cn (Y.-B.Z.); chenming02@caas.cn (M.C.); mayouzhi@caas.cn (Y.-Z.M.); 3National Engineering Laboratory for Wheat and Maize/Key Laboratory of Wheat Biology and Genetic Improvement, Crop Research Institute, Shandong Academy of Agricultural Sciences, Jinan 250100, China; cxytvs@163.com; 4College of Agronomy, College of Agriculture, Yangtze University, Jingzhou 434025, China; fangzhengwu88@163.com

**Keywords:** calcium-dependent protein kinase, abiotic stresses, responsive mechanism, soybean hairy root, *Arabidopsis*

## Abstract

Plants have a series of response mechanisms to adapt when they are subjected to external stress. Calcium-dependent protein kinases (CDPKs) in plants function against a variety of abiotic stresses. We screened 17 CDPKs from drought- and salt-induced soybean transcriptome sequences. The phylogenetic tree divided CDPKs of rice, *Arabidopsis* and soybean into five groups (I–V). *Cis*-acting element analysis showed that the 17 CDPKs contained some elements associated with drought and salt stresses. Quantitative real-time PCR (qRT-PCR) analysis indicated that the 17 CDPKs were responsive after different degrees of induction under drought and salt stresses. *GmCDPK3* was selected as a further research target due to its high relative expression. The subcellular localization experiment showed that GmCDPK3 was located on the membrane of *Arabidopsis* mesophyll protoplasts. Overexpression of *GmCDPK3* improved drought and salt resistance in *Arabidopsis*. In the soybean hairy roots experiment, the leaves of *GmCDPK3* hairy roots with RNA interference (*GmCDPK3*-RNAi) soybean lines were more wilted than those of *GmCDPK3* overexpression (*GmCDPK3*-OE) soybean lines after drought and salt stresses. The trypan blue staining experiment further confirmed that cell membrane damage of *GmCDPK3*-RNAi soybean leaves was more severe than in *GmCDPK3*-OE soybean lines. In addition, proline (Pro) and chlorophyll contents were increased and malondialdehyde (MDA) content was decreased in *GmCDPK3*-OE soybean lines. On the contrary, *GmCDPK3*-RNAi soybean lines had decreased Pro and chlorophyll content and increased MDA. The results indicate that *GmCDPK3* is essential in resisting drought and salt stresses.

## 1. Introduction

Plants inevitably experience a variety of abiotic stresses during their growth and development process, such as drought, salt, and extreme temperature [1]. To adapt to certain environmental changes, plants evolved regulatory pathways in response to different stress signal stimuli [2]. As signals are transmitted through the cell, the plant will regulate the expression of some genes and produce new proteins, which will also change noticeably at the morphological, physiological, and biochemical levels [3].

Calcium ions (Ca^2+^) are ubiquitous and are the most important messenger molecules in higher plant cells. Ca^2+^ is involved in regulating growth and development of plants and functions directly in the biotic and abiotic stress response of plants [4]. When plants are exposed to different external stimuli, Ca^2+^ acts as the second messenger in the cell, and its concentration in the cytoplasm undergoes transient and complex changes, producing Ca^2+^ oscillations, which are transmitted to the calcium-binding protein via calcium receptors, and are then transmitted and amplified by calcium-binding proteins. The signal goes downstream, causing changes in the expression of the corresponding gene, thereby regulating the response of the plant to adverse stimuli [4,5].

Calcium-dependent protein kinases (CDPKs) are a class of typical Ser/Thr protein kinases [6]. CDPKs are widely distributed in some plants, algae, and some lower animals, but not in bacteria, fungi, or higher animals [7,8]. CDPKs are essential in Ca^2+^-mediated signal transduction in plant response to stresses. Unlike other calcium binding proteins, CDPKs are a single peptide chain. CDPKs possess four specific domains: the N-terminal domain, the ATP-binding kinase domain, the autoinhibitory junction domain, and the C-terminus domain [9,10]. The C-terminus domain of CDPKs binds with Ca^2+^, resulting in conformational changes of CDPK proteins and causing activation of the kinase domain and autoimmune changes. This change results in important functions of CDPKs in plant responses to abiotic stresses [11].

CDPKs are widely present in various organs of plants and are expressed in roots, stems, leaves, flowers, fruits, and seeds [12,13,14,15]. CDPKs are mainly involved in physiological activities such as plant drought resistance, salt resistance, disease resistance, hormone signal transduction, photoperiod regulation, and nutrient metabolism [16]. In the past 10 years, research on plant CDPKs developed rapidly, and a large number of CDPKs were found in many plants. A total of 34 CDPKs have been found in *Arabidopsis* [10], 31 CDPKs in rice [11], 40 CDPKs in maize and 20 CDPKs in wheat [17]. In addition, CDPKs have been increasingly studied in horticultural crops, and 19 CDPKs have been identified in cucumber [13], 29 CDPKs in tomato [18], 31 CDPKs in pepper [19], 19 CDPKs in grape [20], and 29 CDPKs in millet [16].

Soybean (*Glycine max*) is one of the most important crops in the world, providing oil and protein to humans and livestock. There are currently 50 CDPKs identified from soybean [21]. The *GmCDPKs* transcript levels change after wounding, exhibiting specific expression patterns after being simulated by *Spodoptera exigua* feeding or soybean aphid (*Aphis glycines*) herbivory, and are largely independent of the phytohormones jasmonic acid and salicylic acid [15]. Based on previous reports, this study screened GmCDPKs with high expression levels in soybean transcriptome sequences from our laboratory following drought and salt stresses, and screened for 17 GmCDPKs responding to drought and salt. The genes were further verified by qRT-PCR analysis, and *GmCDPK3* was selected for further research. The results showed that in transgenic *Arabidopsis*, *GmCDPK3* gene was resistant to drought and salt environmental stress, and this characteristic was verified in soybean root experiment. The discovery could help develop plants that are resistant to drought and salt.

## 2. Results

### 2.1. Phylogenetic Tree Inference in CDPKs

All amino acid sequences were used to create phylogenetic tress in MEGA X 10.0 (Figure 1). The adjacent junction is the complete sequence of CDPKs in *Arabidopsis*, rice, and soybean, and the phylogenetic tree was divided into five groups (I–V). For 17 screened genes, 6 were in group I, 3 in group II, 2 in group III, 5 in group IV, and 1 in group V. 

### 2.2. Analysis of Gene and Protein Structure of 17 Selected GmCDPK

A large number of CDPKs were found in the soybean drought and salt transcriptome. To further analyze their roles in abiotic stress, we selected 17 GmCDPKs and queried basic information in the Phytozome, Pfam, and SMART databases (Appendix A) [22]. The chromosomal locations of the 17 GmCDPK genes (Figure 2) on the 12 chromosomes. As shown in figure, three CDPK genes were mapped on chromosomal 2 and 5; tow CDPK genes on chromosomal 10; one CDPK gene was mapped on chromosomal 1, 3, 4, 6, 11, 14, 16, 18, and 19. To characterize the 17 GmCDPKs, we used the gene structure display server (GSDS) website (http://gsds.cbi.pku.edu.cn/) to analyze genomic sequences by submitting coding DNA. The results showed that 17 GmCDPKs contained an S-TKc protein domain and four EF hand-shaped structures (Figure 3a). The genes had exon-intron structure (Figure 3c). In addition, the 17 GmCDPKs contained nine motifs, in which *GmCDPK*1/13/15 had the same structure, and the structures of other genes were similar (Figure 3b). Results showed that CDPKs tend to have a close genetic relationship with similar structures, suggesting that they evolved from the same pattern.

### 2.3. GmCDPK Protein Tertiary Structure Homology Modeling

To visualize the conserved Ser/Thr protein kinase domain and EF hand structure on the GmCDPK, protein structure models of these 17 GmCDPKs were constructed (Appendix A). The simulated three-dimensional structure shows that in addition to the unique coiled Ser/Thr protein kinase domain, there are four EF-hand structures, consisting of four helical region-bubble region-coil regions with Ca^2+^ binding in the middle of the hand structure region.

### 2.4. Expression of 17 GmCDPKs in Different Tissues and at Different Developmental Stages

Gene registration numbers were submitted to SoyBase (http://soybase.org/soyseq/) [23], and quantitatively predicted tissue expression data was obtained for 14 tissues (Appendix A) (details are in Appendix A). The results showed that *GmCDPK6/7/8/9/10* were highly expressed in roots and basal tissues; *GmCDPK14* was highly expressed at all times in soybean; and most of the other genes were slightly expressed or had no expression under stress-free conditions. These results indicated that these genes are essential in the soybean response to stresses.

### 2.5. Promoter Regions of 17 GmCDPKs Contain Various Stress Response Elements

The 2000 bp sequence before the start codon ATG was cloned in these 17 GmCDPK promoters. To investigate the mechanism of response to abiotic stress, the plant *cis*-acting elements of the 17 GmCDPK promoter regions were submitted to PLACE (http://bioinformatics.psb.ugent.be/webtools/plantcare/html/) (Appendix A). Many regulatory factors that respond to drought and salt stresses have been identified, including ABRE, DRE, ERE, GT-1, LTRF, MYB, and W-box elements. All of the genes except *GmCDPK12* have an ERE *Cis*-acting element, of which *GmCDPK9/13* is the most abundant. *GmCDPK3* contains seven ABRE, four EREs, two GT-1s, one LTRE, two MYBs, and four W-box *Cis*-acting elements. This information indicates that 17 GmCDPKs may be involved in abiotic stress responses.

### 2.6. Candidate Genes Involved in Drought and Salt Stresses

To gain insight into the potential functions of these 17 GmCDPKs in plants subjected to abiotic stresses, we detected the expression patterns under drought and salt stresses by qRT-PCR. After the drought treatment, *GmCDPK1/17* reached the highest expression at the 2nd hour, *GmCDPK2/3/15* had the highest expression at the fourth hour, and *GmCDPK8* and *GmCDPK13* had the highest expression at the first and eighth hours, respectively (Figure 4a). After the salt treatment, the expression of *GmCDPK3/8/11/12/13* reached the highest level at the eighth hour. Furthermore, the transcription levels of other genes slowly increased, but the changes were not significant (Figure 4b). These results indicate that the transcription levels of most of the GmCDPKs are affected by drought and salt stresses. *GmCDPK3* apparently responds to a variety of these stresses, so *GmCDPK3* was selected for further study.

### 2.7. GmCDPK3 Localized on the Cell Membrane 

To determine the subcellular localization of *GmCDPK3*, the *GmCDPK3* gene sequence was fused with the N-terminal end of the GFP reporter gene, and connected to the 16318 hGFP expression vector under the control of cauliflower mosaic virus (CaMV). The constructed vector was transferred to the protoplast of *Arabidopsis* and the location of GFP expression was observed. We found that GmCDPK3-GFP fusion protein was mainly located in the cell membrane (Figure 5).

### 2.8. GmCDPK3 Conferred Drought Tolerance in Transgenic Arabidopsis

Genes have potential functions in enhancing abiotic stress tolerance, particularly in overexpression plants [24,25,26,27,28]. To further study the biological function of *GmCDPK3*, overexpression (OE, in lines 5, 14, and 15) lines were treated with polyethylene glycol (PEG6000). To determine germination rate, seeds of OE and WT were sown on 1/2 MS0 medium containing different concentrations of PEG6000, and at 0, 12, 24, 36, 48, 60, and 72 h, and the number of germinations was recorded. The germination rate on 1/2 MS0 medium did not differ, however, in the medium containing PEG6000, the germination of WT seeds was inhibited to a greater extent than that of OE seeds (Figure 6a). Under normal conditions, the germination rates of WT and OE ranged from 96–100% at 84 h (Figure 6b). Under 9% PEG6000 treatment, the germination rate of OE was 92.79–95.92%, higher than WT (85.71%) (Figure 6c). Under 12% PEG6000 treatment, the germination rate of OE (83.67–96.94%) was still higher than that of WT (73.47%) (Figure 6d). 

Six-day-old *Arabidopsis* seedlings were transferred to 1/2 MS0 medium containing different concentrations of PEG6000 for a week (Figure 7a). Under normal conditions and 6% PEG6000, the root length phenotype of OE was similar to WT, with no differences among lines (Figure 7b, c). Under treatment of 9% and 12% PEG6000, the root length of the OE was significantly longer than WT (Figure 7d, e). With the increase of PEG6000 concentration, the root length became shorter.

In the seedling treatment, 3-week-old OE and WT seedings were subjected to drought for 14 days, and then rehydration for 3 days, and the survival rates were recorded separately. The survival rate of OE after 3 days of fluid replacement was 90.05–91.63%, which was significantly higher than the WT (50.50%) (Figure 10a,c).

### 2.9. Salt Tolerance of GmCDPK3 in Arabidopsis 

To elucidate the role of *GmCDPK3* in plant growth and development under salt treatment conditions, salt tolerance experiments of OE and WT were performed. For the germination assay, seeds of OE and WT were cultured on 1/2 MS0 medium containing different concentrations of NaCl, and the germination rate was measured (Figure 8a). OE and WT had similar germination rates on 1/2 MS0 medium. Under the treatment of 100 mM NaCl, the germination rate of OE seeds was 82.31–85.76%, which was higher than that of WT seeds (72.11%). Under the treatment of 125 mM NaCl, the germination rate of OE seeds was 65.31–86.73%, which was higher than that of WT seeds (61.22%). When treated with 150 mM NaCl, the germination rate of OE seeds (72.45–82.65%) was significantly higher than that of WT seeds (60.20%) (Figure 8b–e). The germination of OE and WT seeds was inhibited under NaCl treatment.

For root length analysis, OE and WT seeds were grown on 1/2 MS0 medium at 22 °C for a week, then transferred to 1/2 MS0 medium containing different concentrations of NaCl and grown for 7 days (Figure 9a). Under normal conditions, OE and WT had similar root length. The OE roots were significantly longer than WT under 100 mM NaCl treatment. Under the treatment of 125 mM and 150 mM NaCl, the root length of the OE was significantly longer than WT (Figure 9b–e). With the increase of NaCl concentration, the taproot length became shorter and the number of lateral roots increased in OE plants.

In the seedling treatment, 3-week-old OE and WT seedlings were cultured for 14 days at 250 mM NaCl, and the survival rate of the OE was 91.56–92.36%, which was significantly higher than WT (74.94%) (Figure 10b,d). The results indicate that *GmCDPK3* can be used to increase the tolerance of transgenic plants to salt stress.

### 2.10. Positive Effect of GmCDPK3 in Transgenic Soybean Hairy Roots Under Drought and Salt Treatment 

*GmCDPK3* has a positive effect on drought and salt stresses of transgenic soybean hairy roots. To further investigate the biological function of *GmCDPK3* under drought and salt treatment conditions, we used *Agrobacterium rhizogenes*-mediated transformation of soybean hairy roots to produce *GmCDPK3*-OE (*GmCDPK3* overexpression) and Gm*CDPK3*-RNAi (*GmCDPK3* with RNA interference) soybean hairy roots. In the drought treatment, the hairy roots of the seedlings were dried for 7 days first and then rehydrated for 3 days. Under drought stress, the leaf wilting degree of *GmCDPK3*-RNAi was higher than that of EV (empty-vector) and *GmCDPK3*-OE (Figure 11a). Under salt stress, the phenotype of the plants was observed to be consistent with the drought treatment, in which the *GmCDPK3-*RNAi plants showed more pronounced leaf drooping, yellowing, and wilting (Figure 12a).

To investigate the potential physiological mechanisms by which *GmCDPK3*-OE enhances plant tolerance, we determined proline (Pro), malondialdehyde (MDA) and chlorophyll content in *GmCDPK3*-OE, EV, and *GmCDPK3*-RNAi plants under normal and stress conditions. The results showed that under drought conditions, the MDA content of the *GmCDPK3*-RNAi (63.76 nmol/g) was higher than that of EV (48.41 nmol/g); MDA content (35.42 nmol/g) of *GmCDPK3-OE* was lower than EV (48.41 nmol/g). The Pro and chlorophyll content of *GmCDPK3*-OE (5.02 μg/g and 0.64 mg/g, respectively) were higher than EV (4.88 μg/g and 0.31 mg/g, respectively) and *GmCDPK3*-RNAi (4.1 μg/g and 0.04 mg/g, respectively) (Figure 11b,c). Under salt treatment conditions, the MDA content of the *GmCDPK3*-RNAi (54.57 nmol/g) was significantly higher than EV (41.44 nmol/g) and *GmCDPK3*-OE (31.90 nmol/g). The Pro and chlorophyll content of *GmCDPK3*-RNAi (1.43 μg/g and 0.55 mg/g, respectively) were significantly lower than EV (1.91 μg/g and 0.77 mg/g, respectively) and *GmCDPK3*-OE (4.42 μg/g and 0.83 mg/g, respectively) (Figure 12b,c) (Appendix A).

When cells are damaged or die, trypan blue penetrates the denatured cell membrane and binds to the disintegrated DNA to color the cell, while living cells prevent the dye from entering the cell. Staining of the soybean leaves after salt and drought treatment was observed. Compared with the control, the leaves of the treated group had different degrees of cell membrane damage, and the *GmCDPK3*-OE had the least damage, followed by EV and *GmCDPK3*-RNAi (Appendix A).

## 3. Discussion

In this study, we analyzed 17 GmCDPK basic molecular characterization and identified the *GmCDPK3* gene from soybean. Subcellular localization results suggested that *GmCDPK3* functions in cell membranes. We obtained transgenic *Arabidopsis* seedlings and soybean hairy roots for studying the functions of *GmCDPK3*. Our results indicated that overexpression of *GmCDPK3* improved plants tolerance to drought and salt stresses compared with control, after PEG6000 and NaCl treatments.

The regulation of CDPKs in the response to plant stresses has been widely reported. Overexpression of *SiCDPK24* in *Arabidopsis* enhanced drought resistance [16]. *OsCPK24* overexpressing plants have stronger resistance to low temperature by increasing amino acid content and increasing the GSH/GSSG (reduced glutathione/oxidized glutathione) ratio [23]. Overexpression of rice *OsCPK12* can inhibit the accumulation of intracellular ROS, thereby enhancing the tolerance of transgenic rice to salt stress [24]. *AtCPK10* functions in abscisic acid- (ABA) and Ca^2+^-mediated stomatal regulation in response to drought stress [25]. *AtCPK32*, through phosphorylating ABA-induced transcription factor ABF4, is involved in ABA/stress responses [26]. *OsCPK21* phosphorylates *OsGF14e* to facilitate the response to ABA and salt stress [27]. In our research, qRT-PCR analysis indicated that most of 17 GmCDPKs responded highly to drought or salt treatment. We hypothesize that soybean CDPKs may be play important roles in enhancing tolerance to drought or salt stresses.

Drought, salinity, and other abiotic stresses are important factors that restrict plant growth and affect plant morphology [28]. There are many microorganisms living around the root system of plants. Changing the composition of plant-related flora in the root region and selecting a combination that is adapted to abiotic stress can improve plant resistance to stress sources, promote plant health and drought tolerance. An important factor in alleviating plant drought stress is *mycorrhizal fungi* (PGPR), which promotes plant growth [29]. Studies have shown that promoting the growth of plant *rhizobacteria* can improve the drought resistance of crops [30]. In addition, plants have evolved various physiological and biochemical mechanisms to adapt to environmental changes [28]. Under drought stress, the water loss rate of the drought-sensitive mutant dsm2 was faster than that of the wild type; the photosynthetic rate, biomass and grain yield of the mutant were significantly reduced, while the malondialdehyde level and stomatal aperture increased. *DSM2* gene plays an important role in the regulation of the rice lutein cycle and ABA synthesis; that is, it provides the ability for the establishment of a drought tolerance mechanism in rice [31]. GmNHX1, located in the vacuole membrane, can enhance the salt tolerance of plants by maintaining a high K^+^/Na^+^ ratio and inducing the expression of SKOR, SOS1, and AKT1. Many physiological indicators are often used to verify plant tolerance during growth [32]. It has been proven that proline accumulation under adverse conditions can resist the damage abiotic stress causes plants [33,34]. The higher the content of proline, the stronger the resistance of plants. Malondialdehyde is produced by peroxidation of membrane lipids in tissues or organs when plants are aging or injured in adverse conditions [35]. The higher the malondialdehyde content, the more damaged the plant is. Chlorophyll content can reflect the plant is being hurt by adversity stress level. Research has shown that PSII of salinity–alkalinity stress is the most sensitive; the chloroplast is one of the most important organelles in plant response to salt stress [36]. According to the data obtained in our experiment, it is certain that overexpression of *GmCDPK3* gene can protect plants from adverse conditions.

Roots are one of the main vegetative organs of plants. They absorb water and minerals dissolved in the soil and transport them to the aerial parts of plants [37]. In the case of drought and high salt conditions, in order to improve plant vigor, the roots face the problem of controlling ion migration into or out of the cell membrane to maintain ion balance. In the root length experiment of *Arabidopsis*, it was observed that the root phenotype of *Arabidopsis* gradually changed with the increase of PEG6000 concentration (Figure 7). In the WT, the primary roots gradually shortened, the lateral roots decreased, the length became shorter and the root hairs became sparse. However, in overexpressing lines, with the increase of PEG6000 concentration, *Arabidopsis* showed short roots, increased the number of lateral roots and density of root hair. Under salt treatment conditions, the roots of WT changed in the same way as the drought treatment with the increase of salt concentration (Figure 9). In the overexpressed lines, with the increase of salt concentration, the same primary roots of *Arabidopsis* shortened and the number of lateral roots increased. The number of root hairs increased, but the overall change was not as obvious as the phenotype during drought treatment. Therefore, we hypothesize that CDPK regulates plant adaptation to the stress environment by regulating the multiple pathways of Ca^2+^ in *Arabidopsis* and may have direct or indirect regulatory functions on plant root growth and development.

## 4. Materials and Methods

### 4.1. Phylogenetic Tree Analysis and Gene Source

CDPK genes and protein sequences of *Arabidopsis*, rice, and soybean were obtained from TAIR (http://arabidopsis.org), RGAP (http://rice.plantbiology.msu.edu), and Phytozome (http://megasoftware.net). The phylogenetic tree was inferred using the neighbor-joining method in MEGA 10.0. Based on previously published information [22], combined with the existing soybean drought and salt transcriptome data in the laboratory to screen out all CDPK genes, and selected 17 genes with a multiple log2 Fold Change greater than 2 for the following experiment. 

### 4.2. Sequence Analysis of 17 Drought and Salt-Tolerant GmCDPKs

Gene sequences of 17 GmCDPKs were analyzed at the transcriptome level, and the domain and tertiary structure of 17 GmCDPKs were separately analyzed at the protein level. To depict the chromosomal locations of the 17 GmCDPK genes Map Gene 2 Chromosomal (http://mg2c.iask.in/mg2c_v2.0/) was used. The tertiary structure was predicted using SWISS-MODEL (https://www.swissmodel.expasy.org), and evaluated in SAVES (https://servicesn.mbi.ucla.edu/SAVES/) [38]. The SMART (http://smart.embl-heidelberg.de/) online tool and ExPASy Proteomics Sever (http://expasy.org/) were used to analyze the functional domain of GmCDPKs [39]. The motifs of GmCDPKs were analyzed with the online tool MAMA (https://www.ebi.ac.uk/Tools) [40]. An exon–intron structure map was created using GSDS online tools (http://gsds.cbi.pku.edu.cn/). Expression of 17 GmCDPKs at different tissue and developmental stages was analyzed by SoyBase (https://www.soybase.org/soyseq/) [41].

### 4.3. Promoter Analysis of GmCDPKs

The 2000 bp region upstream of the ATG start codon of the 17 GmCDPKs was submitted to PLACE (http://bioinformatics.psb.ugent.be/webtools/plantcare/html/) to identify the cis-acting elements and calculate the number of each element [42]. 

### 4.4. Planting of Plant Materials 

Soybean seeds (Our Laboratory supply, Tiefeng 8) were planted in pots until the seedlings grew to 10 cm and had two new leaves, and then the seedlings were subjected to drought and salt stresses. Drought stress involved putting soybean seedlings on filter paper for rapid drought for 0, 1, 2, 4, 8, 12, 24, and 48 h; salt treatment involved transferring seedlings to 250 mM NaCl with the same timing, samples were separately taken and immediately immersed in liquid nitrogen and stored at −80 °C for RNA extraction [43].

The seeds of *Arabidopsis* (Col-0) were sterilized and planted on 1/2 MS0 medium, after vernalization at 4 °C for 3 days. Plates containing seeds were placed at 22 °C and in light conditions of 40 μmol/m^2^/s^1^ with a photoperiod of 16h light/8h dark in the growth chamber the seedlings were then used in further experiments [43,44].

### 4.5. RNA Extraction and qRT-PCR

Plant samples stored at −80 °C were run according to the method provided in the Plant Total RNA Extraction Kit (TIANGEN). qRT-PCR was performed using the experimental method provided by the PrimeScript^TM^ RT Kit (Takara, Shiga, Japan). The primers were designed by Primer Premier 5.0, in which the soybean actin gene was used as a control. An ABI Prism 7500 real-time PCR system (Applied Biosystems, Foster City, CA, USA) was used to perform qRT-PCR [26]. The resulting data was analyzed using the 2^−ΔΔCT^ method [45].

### 4.6. Subcellular Localization of GmCDPK3

The full-length cDNA sequence of *GmCDPK3* was ligated to the N-terminus of the hGFP gene carrying the CaMV 35S promoter. The recombinant plasmid of *GmCDPK3*-GFP was transformed into *Arabidopsis* protoplasts using a PEG4000-mediated method [46]. It was placed in the dark for more than 12 h and the GFP signal was detected by laser scanning confocal microscopy (Zeiss LSM 700, Oberkochen, Germany) [44].

### 4.7. Drought and Salt Stress Assays of Transgenic Arabidopsis Plants

The full-length cDNA sequence of *GmCDPK3* was transformed into a pCAMBIA1302 plant transformation vector to obtain OE, and then transformed into *Agrobacterium tumefaciens* (GV3101) after sequencing. The gene of interest was transferred into WT *Arabidopsis* (Col-0) plants using the floral dip method mediated by *Agrobacterium* [47]. The seeds were sterilized with 70% alcohol and 0.1% sodium hypochlorite. After vernalization for 3 days at 4 °C, the plates containing the seeds were transferred to a growth chamber [46]. The qRT-PCR analysis of *GmCDPK3* gene expression was conducted in 3-week-old *Arabidopsis* seedlings (Appendix A).

For germination rate experiments, sterilized seeds from WT and OE plants (lines 5, 14, and 15) were sown in various concentrations of PEG6000 (0, 6, 9, and 12%) or NaCl (0, 100, 125, and 150 mM). Each concentration was repeated three times on 1/2 MS0 medium. The plate was housed in a growth chamber maintained at 22 °C, with illumination intensity of 40 μmol/m^2^/s^1^ and photoperiod of 16h-light/8h-dark [48,49]. The number of germinating seeds was counted every 12 h and at least 80 seeds per genotype were measured, and each experiment was repeated three times.

For root growth experiments, sterilized WT and OE seeds were seeded on 1/2 MS0 medium. After 5 days, the seedlings were transferred to medium containing different concentrations of PEG6000 (0, 6, 9, and 12%) or NaCl (0, 100, 125, and 150 mM) for one week, and each concentration was repeated three times. Photographs were taken one week later and the root length was assessed by the Epson Expression 11000XL Root Scan Analyzer (Epson, Nagano, Japan) [44]. To test drought and salt response at later developmental stages, 3-week-old seedlings were treated with drought or 250 mM NaCl for 14 days. The experiment was repeated three times for each concentration and treatment. In the end, the survival rate was calculated, and seeds were photographed.

### 4.8. Vector Construction of GmCDPK3

After amplification of the CDS of *GmCDPK3*, the restriction site sequences (NcoI and BsTEII) and gene-specific primers were ligated to the end of *GmCDPK3*. The PCR products and the pCAMBIA3301 vector were digested with NcoI and BsTEII (ThermoFisher Scientific, USA), respectively, and the product was ligated to obtain pCAMBIA3301-*GmCDPK3* [44,50]. The RNAi sequence was selected on the sense strand encoding the mRNA, the sense strand and the antisense strand of the selected interference sequence were inserted before and after the sequence, and the non-coding sequence (153 bp) from maize was inserted in the middle to form a large hairpin structure. The structure was clipped to cause the gene to be disturbed without expression. After selecting the sequence, it was submitted to the AUGCT company (Beijing, China) for synthesis and testing. The RNAi sequence and the pCAMBIA3301 vector were digested with NcoI and BsTEII, respectively, and the product was ligated to obtain pCAMBIA3301-*GmCDPK3i.*

### 4.9. Transformation of Soybean Hairy Roots

Soybean seeds (Williams 82) were subjected to *Agrobacterium tumefaciens*-mediated transformation of the hairy root [51]. The seeds were planted in pots for 9–14 days, and after the seedlings grew to 10 cm, the hairy root strains carrying the genes of interest were inoculated 2 mm below the soybean cotyledons, and the soil was overturned until the cotyledons were exposed. The newly grown roots were treated after bacterial infection, and the old roots were removed and cultured for 4 days, then subjected to drought and 250 mM NaCl treatment for 16 and 7 days, respectively [52]. Due to the uncertainty of inoculation, the experiment must be expanded to ensure that each treatment is repeated at least three times. The qRT-PCR analysis of *GmCDPK3* expression in *GmCDPK3*-OE, EV-control (3301), and *GmCDPK3*-RNAi transgenic soybean hairy roots before processing (Appendix A).

### 4.10. Trypan Blue Staining

The isolated at least 5 leaves from the same position of soybean seedlings were treated with drought for 7 days, and 250 mM NaCl for 3 days. The leaves were completely immersed in a 0.5% trypan) solution for 12 h, then immersed in 75% ethanol for decolorization until the leaves turned white and were photographed to visualize leaf staining [44].

### 4.11. Determination of Pro, MDA, and Chlorophyll Contents

The leaves were tested for Pro, MDA, and chlorophyll content after 1–3 days of drought or 250 mM NaCl stress [52]. Untreated leaves were used as controls, all measurements were repeated three times and statistical analysis was performed using the ANOVA test. Pro, MDA, and chlorophyll contents were measured in accordance with the instructions provided in the corresponding kit produced by Suzhou Comin Biotechnology Co., Ltd (Suzhou, China).

## 5. Conclusions

We screened GmCDPK genes induced by abiotic stresses from transcriptome, and our results demonsrated that *GmCDPK3* can increase plant resistance to drought and salt stresses.

## Figures and Tables

**Figure 1 ijms-20-05909-f001:**
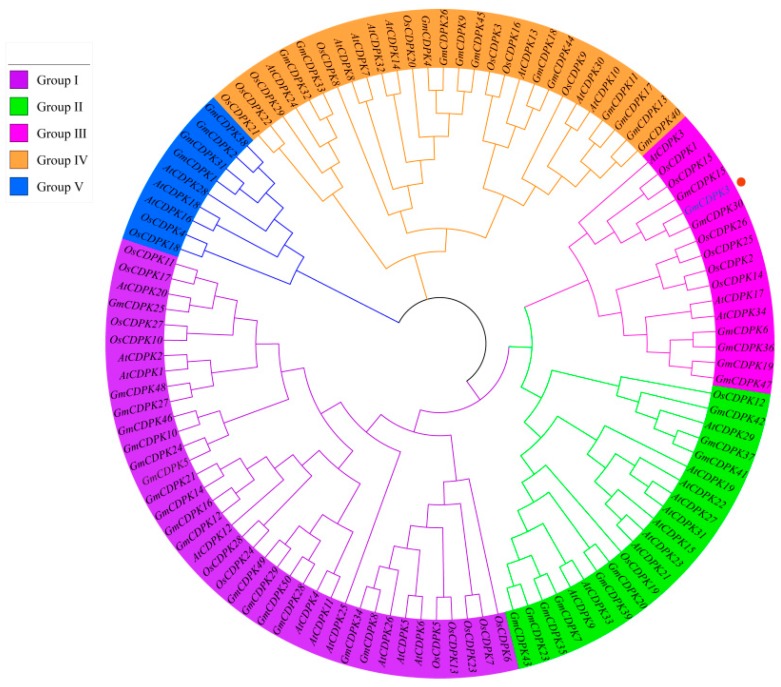
Phylogenetic relationships of CDPKs among *Arabidopsis*, rice, and soybean. The phylogenetic tree was generated by comparison of the CDPK amino acid sequences in MEGA 10.0. The neighbor-joining (NJ) method was used and the bootstrap value was set to 1000. Frequency values (%) higher than 50 are displayed. All the genes were divided into five groups and *GmCDPK3* is classified as group III.

**Figure 2 ijms-20-05909-f002:**
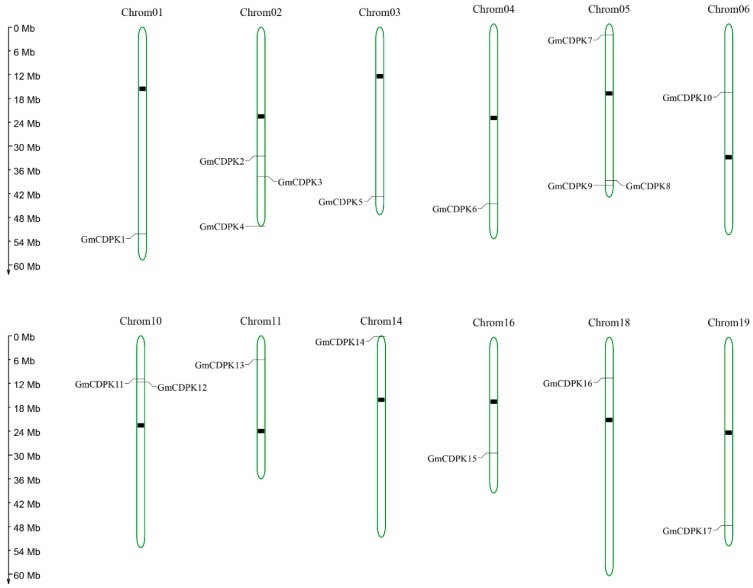
Chromosomal locations of the 17 GmCDPK genes. Using Map Gene 2 Chromosomal to depicted chromosomal location. The location of the gene is marked with short lines. black mark is the centromere of the chromosome. There are 17 GmCDPK genes that are distributed on 12 chromosomes.

**Figure 3 ijms-20-05909-f003:**
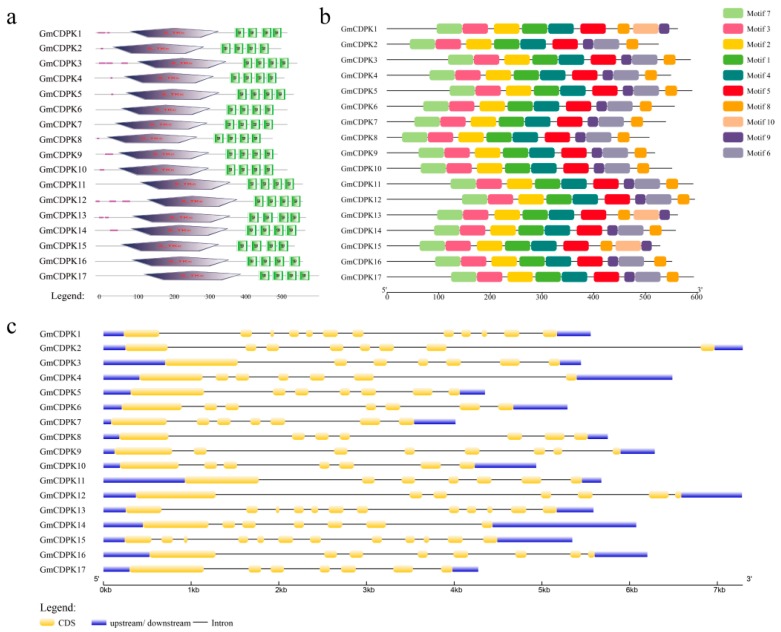
Bioinformatics analysis of 17 GmCDPK proteins and genes. The gene and protein sequences of 17GmCDPK were analyzed with bioanalysis tools. (**a**) The domain of the GmCDPK protein; (**b**) GmCDPKs protein motif; and (**c**) intron exon structure of GmCDPKs.

**Figure 4 ijms-20-05909-f004:**
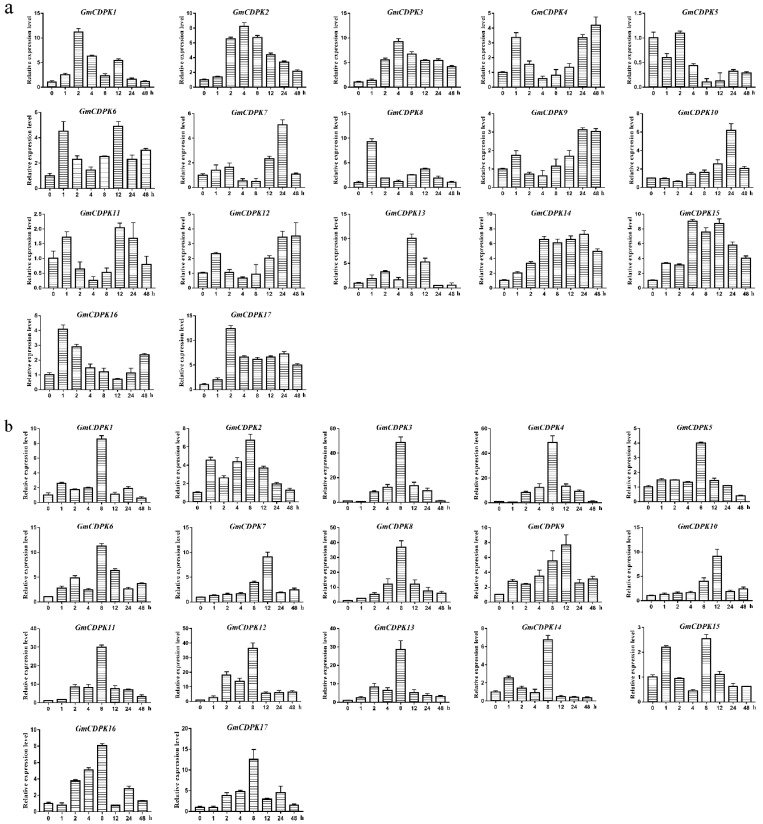
Expression patterns of the 17 GmCDPKs under drought and salt treatment. Using qRT-PCR to measure expression levels of 17 GmCDPKs under drought and salt stresses. (**a**) Drought treatment was applied for 0, 1, 2, 4, 8, 12, 24, and 48 h. (**b**) Expression levels of 17 GmCDPKs under 250 mM NaCl treatment for 0, 1, 2, 4, 8, 12, 24, and 48 h. The result is the mean ± SD of three experiments.

**Figure 5 ijms-20-05909-f005:**
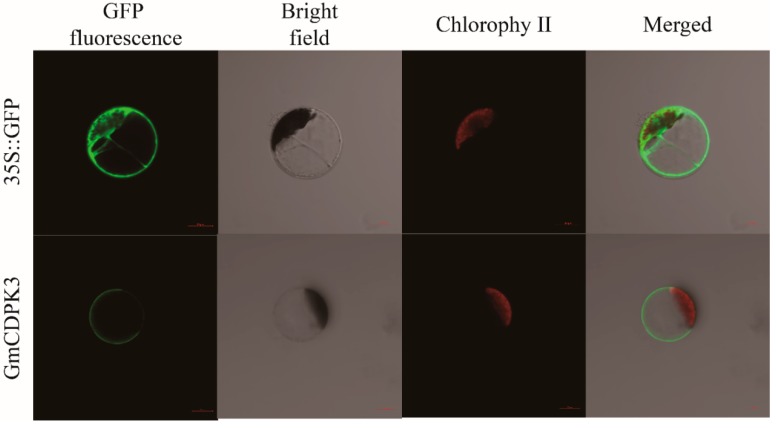
Subcellular localization of *GmCDPK3* overexpression plants in *Arabidopsis*. Plasmid labeled with GFP was transferred into the protoplast of *Arabidopsis* and subcellular localization image of GmCDPK3-GFP was observed under a laser scanning confocal microscope, and the image showed that it was expressed in the cell membrane. Bar = 20 μm.

**Figure 6 ijms-20-05909-f006:**
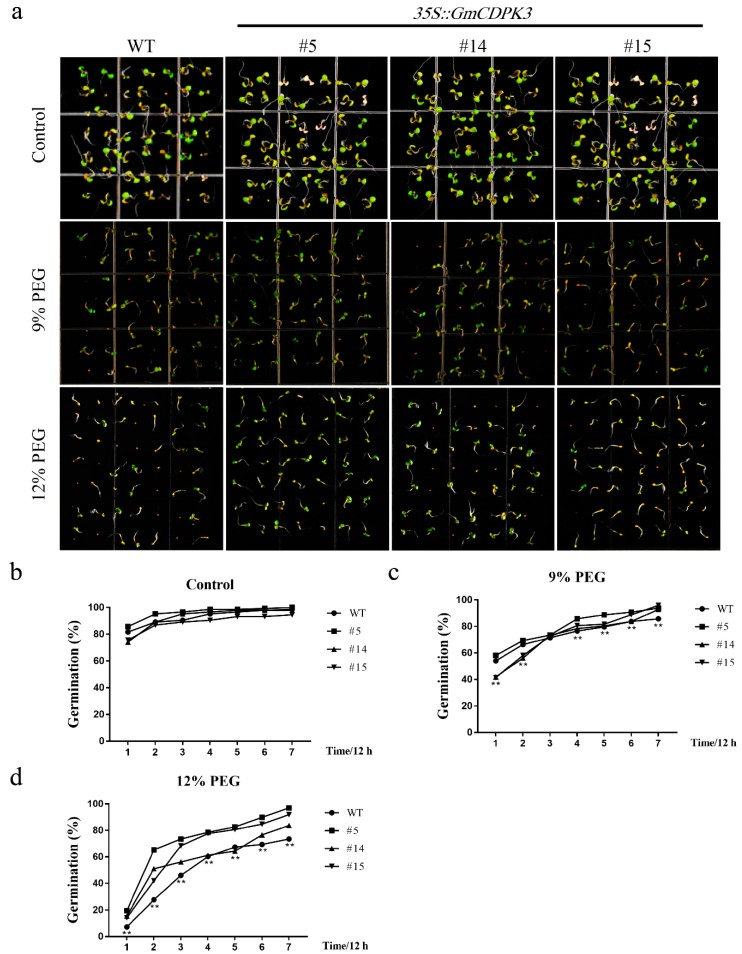
Germination rate of the OE and WT under different concentrations of PEG6000. After disinfection of *Arabidopsis* seeds, select the same strong seeds and sow them on the corresponding medium. Record the number of germinations every 12 h., and each experiment was repeated three times. (**a**) Germination rate phenotypes treated with different concentrations of PEG6000. (**b**) Untreated control germination rate statistics. (**c**) Germination rate statistics under 9% PEG treatment. (**d**) Germination rate statistics under 12% PEG treatment. Data are mean ± SD of three experiments (*n* = 64). The ANOVA test showed a significant difference (** *p* < 0.01).

**Figure 7 ijms-20-05909-f007:**
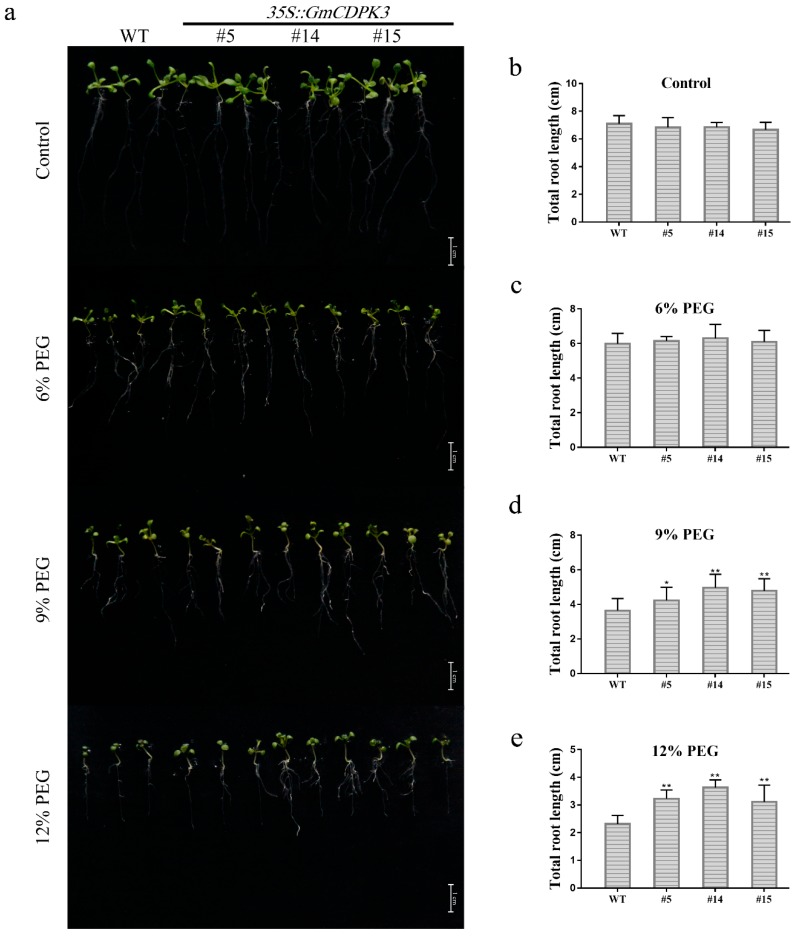
Root length phenotype of the OE and WT under different concentrations of PEG6000. Seedlings with consistent root length and growth state were selected to grow in Ms0 medium for about three days and transferred to another medium to observe root length. Each experiment was repeated three times. (**a**) Root growth phenotypes of OE and WT under different concentrations of PEG6000. (**b**) Untreated control total root length statistics. (**c**) Total root length statistics under 6% PEG treatment. (**d**) Total root length statistics under 9% PEG treatment. (**e**) Total root length statistics under 12% PEG treatment. Data are mean ± SD of three experiments (*n* = 30). The ANOVA test showed a significant difference (* *p* < 0.05, ** *p* < 0.01).

**Figure 8 ijms-20-05909-f008:**
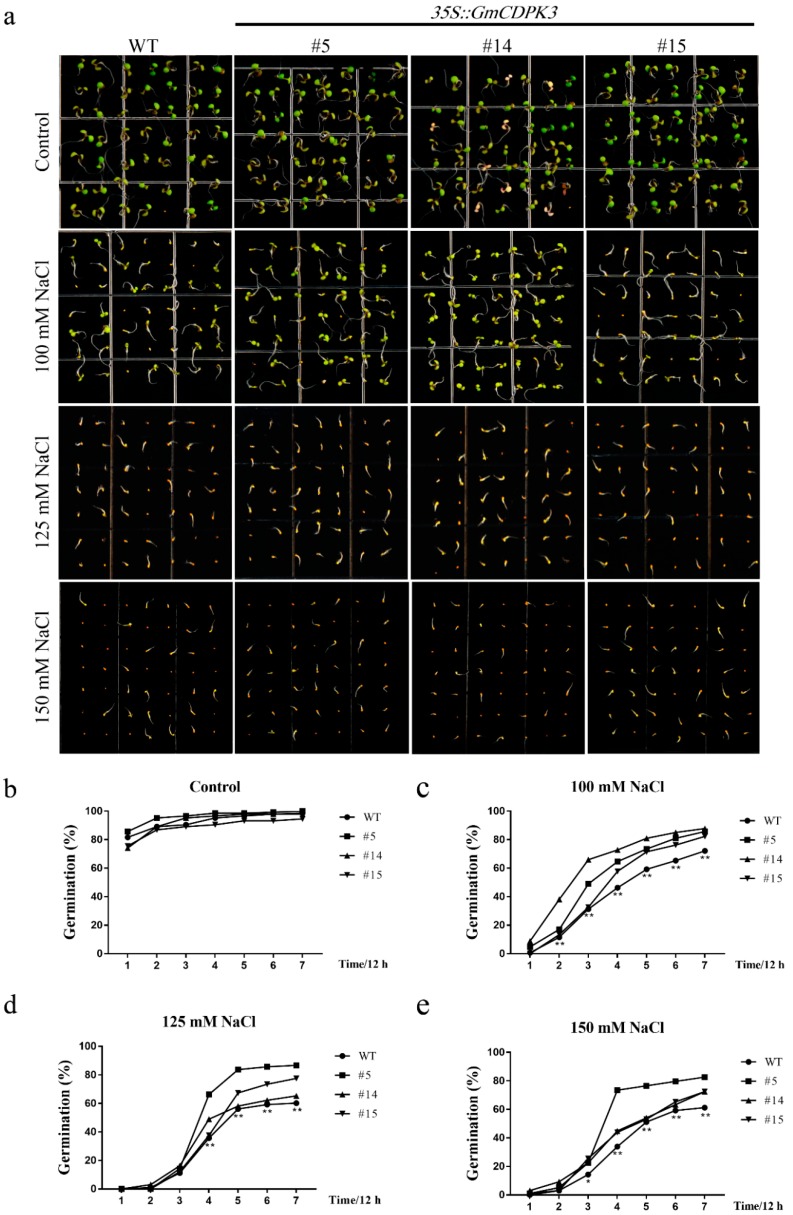
Germination rate of OE and WT under different concentrations of NaCl. After disinfection of *Arabidopsis* seeds, select the same strong seeds and sow them on the corresponding medium. Record the number of germinations every 12 h., and each experiment was repeated three times. (**a**) Germination rate phenotypes treated with different concentrations of NaCl. (**b**) Untreated control germination rate statistics. (**c**) Germination rate statistics under 100 mM NaCl treatment. (**d**) Germination rate statistics under 125 mM NaCl treatment. (**e**) Germination rate statistics under 150 mM NaCl treatment. Data are mean ± SD of three experiments (*n* = 64). The ANOVA test showed a significant difference (* *p* < 0.05, ** *p* < 0.01).

**Figure 9 ijms-20-05909-f009:**
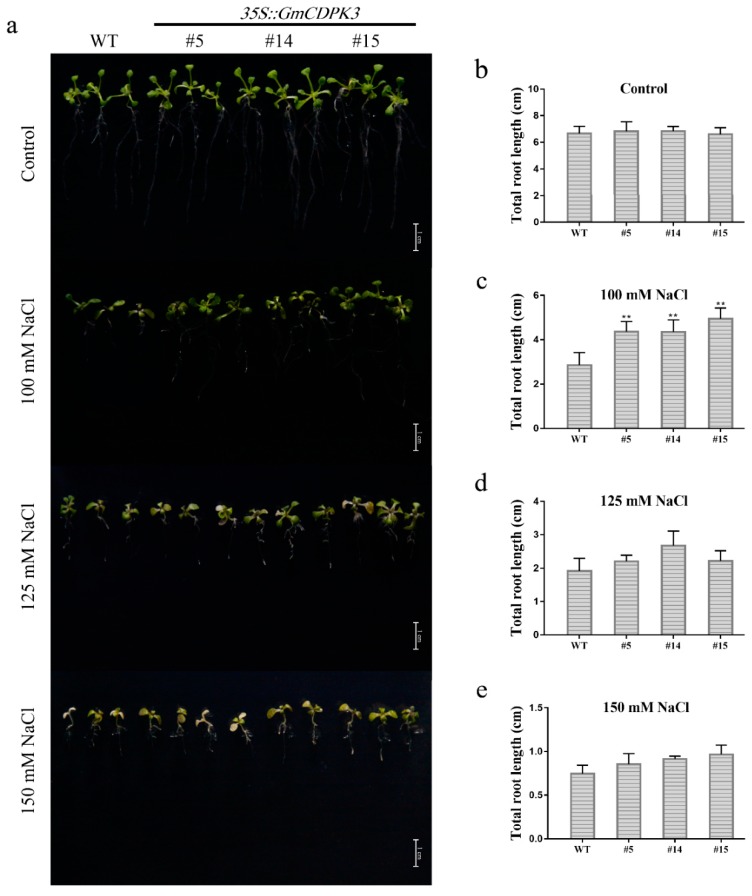
Root length phenotypes of OE and WT under different concentrations of NaCl. Seedlings with consistent root length and growth state were selected to grow in Ms0 medium for about three days and transferred to other medium to observe root length. Each experiment was repeated three times. (**a**) Root growth phenotypes of OE and WT under different concentrations of NaCl. (**b**) Untreated control total root length statistics. (**c**) Total root length statistics under 100 mM NaCl treatment. (**d**) Total root length statistics under 125 mM NaCl treatment. (**e**) Total root length statistics under 150 mM NaCl treatment. Data are mean ± SD of three experiments (*n* = 30). The ANOVA test showed a significant difference (** *p* < 0.01).

**Figure 10 ijms-20-05909-f010:**
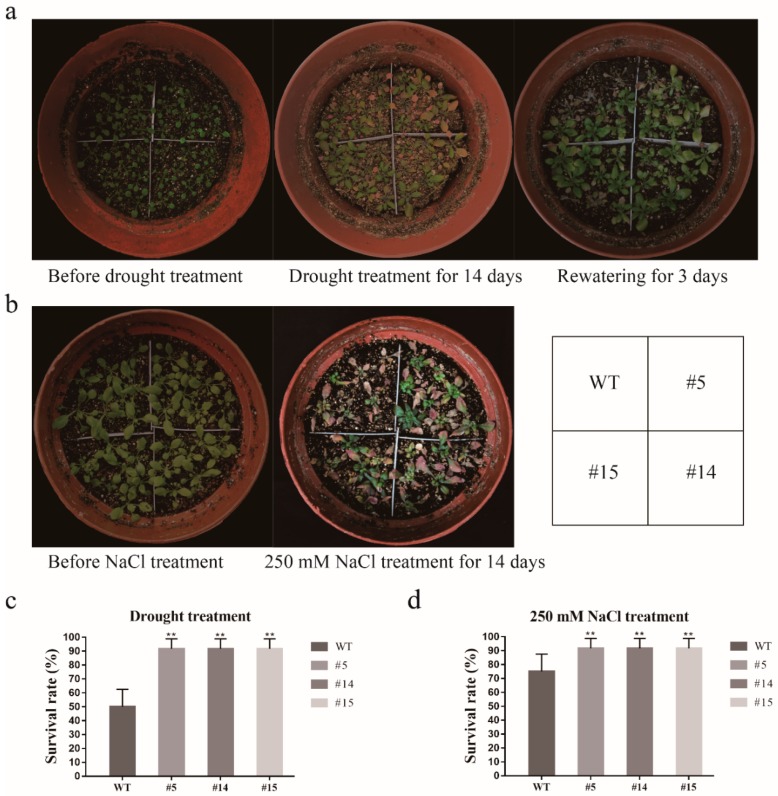
OE and WT phenotypes at the seedling stage under drought and salt treatments. The seedlings of *Arabidopsis* with the same growth condition were transferred to the soil, after growing for two weeks, they were treated with drought and salt. (**a**) Drought-tolerant phenotype of OE and WT in the absence of water and after rehydration. (**b**) Salt-tolerant phenotype of the OE and WT in 250 mM NaCl treatment. (**c**) Monitoring the survival rate of OE and WT under water stress 3 days after rehydration. (**d**) Monitoring the survival rate of OE and WT under 250 mM NaCl treatment for 14 days. Data are means ± SD of three experiments (*n* = 24). The ANOVA test showed a significant difference (** *p* < 0.01).

**Figure 11 ijms-20-05909-f011:**
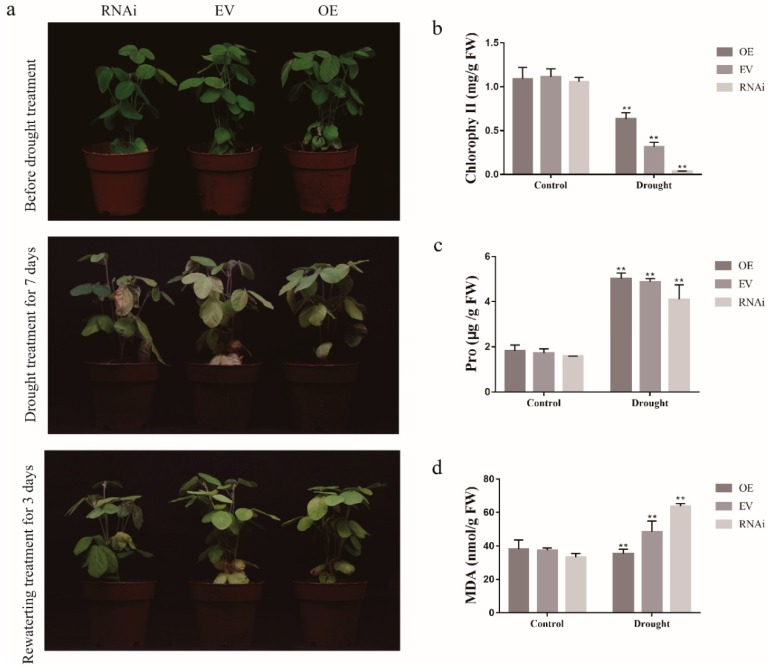
Phenotypic analysis of *GmCDPK3*-OE, *GmCDPK3*-RNAi, and EV under drought stress. Soybean plants with the new root hair were transplanted and grown for another week without watering and drought treatment three times per group. (**a**) Phenotypes of *GmCDPK3*-OE and *GmCDPK3*-RNAi under normal conditions and drought treatments. (**b**) Changes in chlorophyll content in *GmCDPK3-OE*, *GmCDPK3*-RNAi and EV under drought stress. (**c**) Changes in Pro content in *GmCDPK3-OE*, *GmCDPK3*-RNAi, and EV under drought stress. (**d**) Changes in MDA content in *GmCDPK3*-OE, *GmCDPK3*-RNAi, and EV under drought stress (** *p* < 0.01).

**Figure 12 ijms-20-05909-f012:**
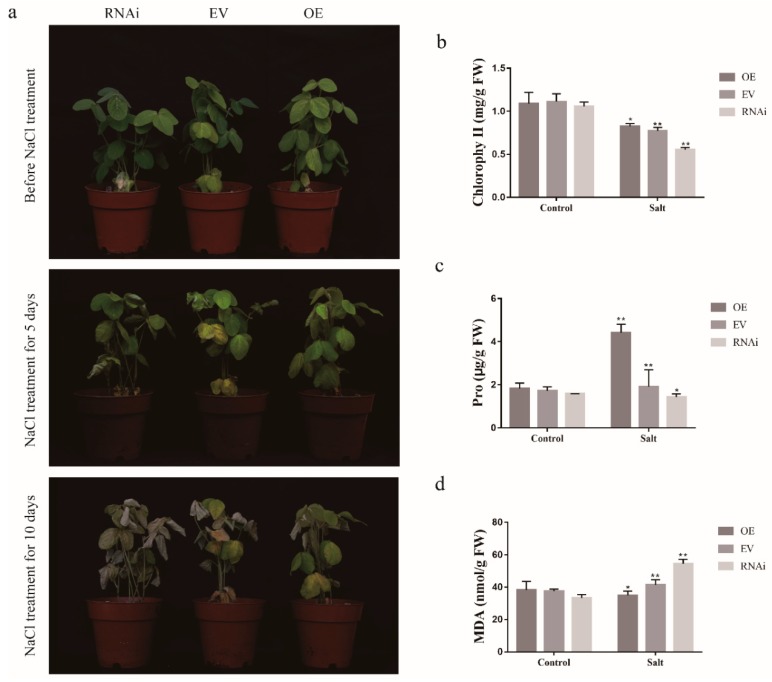
Phenotypic analysis of *GmCDPK3*-OE, *GmCDPK3*-RNAi, and EV under salt stress. Soybean plants with the new root hair were transplanted and grown for another week and 250 mM NaCl treatment three times per group. (**a**) Phenotypes of *GmCDPK3*-OE and *GmCDPK3*-RNAi under normal conditions and salt treatments. (**b**) Changes in chlorophyll content in *GmCDPK3-OE*, *GmCDPK3*-RNAi, and EV under salt stress. (**c**) Changes in Pro content in *GmCDPK3-OE*, *GmCDPK3*-RNAi, and EV under salt stress. (**d**) Changes in MDA content in *GmCDPK3*-OE, *GmCDPK3*-RNAi, and EV under salt stress (* *p* < 0.05 ** *p* < 0.01).

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
