# Peer review of "Functional Analysis of the Soybean *GmCDPK3* Gene Responding to Drought and Salt Stresses"

_ijms, 2019, doi:10.3390/ijms20235909_

Round 1

Reviewer 1 Report

This paper entitled “Functional Analysis of the Soybean GmCDPK3 Gene Responding to Drought and Salt Stresses” describes the role of Soybean GmCDPK3 in drought and salt response by bioinformatics assay, and using Overexpress GmCDPK3 in Arabidopsis and Soybean. The GmCDPK3-OE Soybean showing resistance to drought and salt by less degradation of chlorophyll, more accumulation of Pro, and also less ROS detected in GmCDPK3-OE Soybean. Conversely, the RNAi lines show sensitive to drought and salt. The content is very descriptive (phenotypic alterations, physiological changes). I think the manuscript is well written, and integrate previous studies clearly. The manuscript needs some major revisions prior to being considered for acceptance.

1, I think the manuscript contain too many main figures, some figures can go to the supplementary data, as figure 3 and 4, also table 1 and 2.

2, in figure 14, I can’t detect the difference of Trypan blue stained leaves of GmCDPK3 (OE), GmCDPK3i (RNAi) and EV. Please remove the data or leave it in supplementary part.

3, How to explain the root length of GmCDPK3 overexpress lines in Arabidopsis show much difference to  PEG at 6% and 9%, and to NaCl at 100 mM and 125 mM, but show obviously difference at germination and survival.

4, At the discussion part, The authors didn’t cited the right paper as a link to explain the physiological changes of MDA, chlorophyll, and Pro in drought and NaCl stresses, so I recommend one paper as (Du, H. et al. 2010 Plant Physiol. 154:1304-1318) that have also touched on this subject. The discussion should include these precedent examples and how the present submission fits into the overall context of your MS.

Minor points.

P2 L58  delete ‘Calcium-dependent protein kinases’ before ‘CDPKs’ P2  L44: add ‘process’ after ‘growth and development’ P2  L84: the authors should add some conclusive word at the end of introduction to point out the what’s new in your manuscript. In Figure 2a and figure 5 most words are too small for readers. In figure 7b-7d, you miss some asterisk, In figure 8c,8d, mark the wrong PEG concentration. In figure 9b-9e, and figure 11d,11e delete ‘rate’, it will confuse readers. P8  L164: the add the information of concertation of NaCl

P9  L176: add ‘Overexpression plants’ after ‘GmCDPK3

Author Response

Dear reviewers:

Thank you very much for your earnest work and your valuable advice to me, which is of great help to me.It also helped me refine my manuscript.

The author.

Reviewer 2 Report

The manuscript proposed by Wang et al. is a functional analysis of GmCDPK3 in response to drought and salt stresses.

Using phylogenetic and mRNA expression analysis, they identified GmCDKP3 gene involved in drought and salt stress response. To go further, they characterized the phenotype of overexpression lines in Arabidopsis and Soybean.

As expected, they found that the overexpression of CDKP3 in Arabidopsis or in Soybean induces stress tolerance.

At the beginning of the manuscript, the authors focused their work on 17 GmCDKs. It is not clear for me how the 17GmCDKs were selected. In the manuscript, the authors mentioned previous reports without any references. Without any supporting data, this specific selection is not clear for me and inconsistent.

Additionally, for most experiments, it is unclear if the authors present technical repeats or biological repeats.

Many typos appear in the main body of the manuscript (e.g. lane 72, 181)

Overall, this manuscript presents low interest for a large audience. I would probable expect such a paper in a journal rather oriented to plant physiology.

Author Response

(The authors gave the same response as above.)

Reviewer 3 Report

The manuscript by Wang et al. entitled " Functional Analysis of the Soybean GmCDPK3 Gene Responding to Drought and Salt Stresses" is generally an interesting paper and relevant to the International  Journal of Molecular Sciences. However, I think that manuscript is not suitable for publication in the present form and needs revision.

The paper is generally straightforward, but some sections of manuscript need more attention (including discussion of results and references). The Authors should improve some key words (without title repetition). I think that purpose of the study must be defined by Authors.  In Materials and methods the Authors should add some details on proline, MDA and chlorophyll content measurements. The Authors should also consider the changes in presentation of results - all figures descriptions must longer and contain information on material,  treatment  and used abbreviations (e.g. see Fig. 11, 13-14). I think that Authors should also improve description of results. The detailed  results provided in the text look rather strange (e.g. p.16, l. 268-276). I suggest to prepare adequate tables as supplementary material (and add remark to it in Results).  The Discussion section need more attention - is rather too short - there is more literature on that subject that could be cited and discuss in this section. Additionally, figures could be cited/mentioned in the Discussion, also the Conclusions should be add at the end of the section. The Authors could add more references to the list - there is much more literature on the studied subject. Also the References list must be carefully checked and improved, according to Instruction for Authors. In addition, there are a lot of small mistakes in the text , especially in reference list, that need to be corrected by Authors.

Author Response

(The authors gave the same response as above.)

Round 2

Reviewer 2 Report

I did not notice this information. I think this manuscript is suitable for this special issue. I do not have any comments. I think this manuscript can be accepted.     

Author Response

Dear Reviewer:

Thank you very much for your reply.I have no opinion on your evaluation.I am very willing to listen to all the voices of the outside world, because these are the driving force of my progress.Please don't worry about hitting me. I'm strong.Of course, it is also possible that there are something wrong with my understanding. Anyway, I should express my gratitude to you for your help.

The Author.